# Bioinformatics Analysis and Expression Profiling Under Abiotic Stress of the DREB Gene Family in *Glycyrrhiza uralensis*

**DOI:** 10.3390/ijms26189235

**Published:** 2025-09-22

**Authors:** Linyuan Cheng, Nana Shi, Xiangrong Du, Teng Huang, Yaxin Zhang, Chenjie Zhao, Kun Zhao, Zirun Lin, Denglin Ma, Qiuling Li, Fei Wang, Hua Yao, Haitao Shen

**Affiliations:** 1Key Laboratory of Xinjiang Phytomedicine Resource and Utilization of Ministry of Education, College of Life Sciences, Shihezi University, Shihezi 832003, China; c19109354382@163.com (L.C.); 13209988724@163.com (N.S.); 15680879881@163.com (X.D.); m19996707739@163.com (T.H.); 13565367801@163.com (Y.Z.); 17615627537@163.com (C.Z.); zk2643164423@163.com (K.Z.); 18805092725@163.com (Z.L.); 13040541539@163.com (D.M.); 19996710738@163.com (Q.L.); shtshz-bio@shzu.edu.cn (H.S.); 2Key Laboratory of Oasis Town and Mountain-Basin System Ecology of Xinjiang Production and Construction Corps, Shihezi University, Shihezi 832003, China

**Keywords:** *G. uralensis*, DREB transcription factor, abiotic stress, bioinformatics, gene family, expression analysis

## Abstract

*Glycyrrhiza uralensis* is an important medicinal plant exhibiting strong tolerance to abiotic stresses, including drought and salinity. DREB (Dehydration-Responsive Element-Binding) transcription factors, key members of the AP2/ERF family, play crucial roles in plant growth, development, and stress responses. Based on transcriptome data, we identified 18 DREB transcription factors in *G. uralensis*, designated *GuDREB1* to *GuDREB18*. Bioinformatics analysis revealed genomic sequences ranging from 534 to 2864 bp and coding sequence (CDS) lengths between 525 and 1509 bp. All GuDREB proteins contain a single AP2 domain, including the conserved YRG and RAYD elements, and were predicted to localize to the nucleus. Phylogenetic analysis clustered the *G. uralensis* DREBs with 61 *Arabidopsis thaliana* DREBs into five subgroups, indicating evolutionary conservation. Promoter analysis detected seventeen stress-responsive *cis*-acting elements, encompassing hormone-responsive and abiotic stress-responsive motifs, suggesting diverse biological functions. Tissue-specific expression profiling revealed *GuDREB* transcription in both aerial and underground parts. Drought stress induced varying degrees of *GuDREB* expression, confirming their involvement in stress responses. Notably, *GuDREB10* expression increased significantly in underground parts, while *GuDREB15* showed pronounced upregulation in aerial parts under drought; the *GuDREB15* promoter contained the highest number of light-responsive elements (23), potentially explaining its aerial tissue specificity. Drought stress significantly increased abscisic acid (ABA) content. Underground parts exhibited higher initial sensitivity to drought, whereas aerial parts displayed a more sustained response; ABA levels overall showed an initial increase followed by a decline. This study expands the *G. uralensis* DREB gene database, provides a foundation for selecting stress-resistance genes, and offers insights into DREB functional roles in abiotic stress responses in this key medicinal species.

## 1. Introduction

*G. uralensis*, a species within the genus *Glycyrrhiza Linn*. (Fabaceae, Papilionoideae subfamily), is a significant medicinal plant. Its dried roots and rhizomes are used medicinally, exhibiting diverse pharmacological properties including analgesic, anti-inflammatory, anti-ulcerative, antiviral (particularly against hepatitis), anticancer, and anti-HIV activities [1,2,3]. First documented in the ancient Chinese pharmacopeia Shennong Bencao Jing (Classic of Herbal Medicine), where it was classified as a superior herb, *G. uralensis* is now widely utilized in the food, beverage, and cosmetic industries. Wild *G. uralensis* populations are primarily distributed across the arid and semi-arid regions of Northwest China [4]. This plant demonstrates considerable resilience to environmental challenges, including drought, cold, and salt–alkali stress, and is valued for its role in windbreak and sand fixation, making it an important ecological and economic crop. During evolution and development, plants are frequently exposed to adverse abiotic stresses such as drought, low temperature, and high salinity. Upon encountering such stresses, plants initiate adaptive responses at morphological, physiological, and transcriptional levels to mitigate the detrimental effects.

Abscisic acid (ABA), a key stress hormone, regulates numerous physiological processes involved in plant stress responses. It modulates stomatal closure, maintains tissue water balance, and controls the expression of specific stress-responsive genes [5]. Under drought stress, numerous stress-resistance genes are highly induced. The expression of these genes can be regulated not only through ABA-dependent signaling pathways but also via ABA-independent pathways. Dehydration-Responsive Element Binding proteins (DREBs) play a crucial role in the latter process.

DREB transcription factors specifically bind to the DRE/CRT (C-repeat/Dehydration Responsive Element) cis-acting element and activate the expression of downstream stress-tolerance genes. AP2/ERF represents the largest family of transcription factors in plants, playing vital roles in diverse biological processes including growth, development, stress responses, and hormone signaling [6]. DREB transcription factors belong to a specific subfamily within the AP2/ERF superfamily, characterized by their ability to bind DRE elements [7]. The AP2 domain, their sole conserved domain, comprises only 57–70 conserved amino acid residues. Within this domain, the valine (V) at position 14 and the glutamic acid (E) at position 19 are critical for the binding affinity of the transcription factor to the cis-acting element [8,9]. This domain typically folds into a characteristic three-dimensional configuration of three anti-parallel β-sheets and one α-helix. Okamuro et al. [10] identified two highly conserved motifs within the AP2 domain: a YRG element near the N-terminus, rich in basic and hydrophobic amino acids, which mediates DNA binding activity; and an approximately 40-amino acid RAYD element, which modulates the binding strength of the YRG element to DNA. Furthermore, members of the DREB family are classified into six subgroups (A1 to A6) based on structural and sequence features, with each subgroup exhibiting distinct functions. The DREB subfamily specifically binds to the DRE/CRT cis-acting elements associated with drought and cold responses, thereby inducing the expression of downstream target genes [11,12]. Subgroup A1 members are primarily involved in cold stress responses, while A2 members are key regulators of drought and high-salt stress responses. The functions of A1 and A2 subgroups have been characterized in species such as *Oryza sativa* [13], *Sorghum bicolor* [14], and *Polygonum cuspidatum* [15]. Subgroups A3 to A6 members are implicated in both plant growth and development and responses to abiotic stresses.

To date, DREB family members have been identified in a wide range of plant species, including *Fragaria vesca* [16], *Triticum aestivum* [17], *Brassica napus* [18], *Oryza sativa* [19], and *Solanum tuberosum* [20]. Numerous studies have demonstrated that overexpression of *DREB* genes in *Arabidopsis thaliana* or various crops significantly enhances the transgenic plants’ tolerance to diverse abiotic stresses, such as drought, salinity, and low temperature. Although several transcription factors associated with abiotic stress responses have been identified in *G. uralensis* in recent years, research specifically focusing on DREB transcription factors in this species remains limited. Therefore, the identification of DREB transcription factors in *G. uralensis* and the analysis of their expression patterns under drought stress are crucial. This work will provide an essential theoretical foundation for further elucidating the functional roles of *DREB* genes in the drought stress response of *G. uralensis*. Furthermore, it holds significant potential for guiding transgenic breeding strategies aimed at enhancing stress resistance. Consequently, this study employs transcriptome sequencing and bioinformatic analyses to identify *GuDREB* genes and characterize their tissue-specific expression profiles under drought stress. The findings will lay a theoretical groundwork for screening elite stress-resistance genes. Moreover, this research is of great importance for investigating the stress tolerance mechanisms of *G. uralensis* in adverse environments, breeding novel stress-resistant crop varieties, and ultimately improving crop yield.

## 2. Results

### 2.1. Impact of Drought Stress on ABA Accumulation in G. uralensis Tissues

Bulleted lists look like this: Drought stress significantly increased ABA content in both underground and aerial parts of *G. uralensis* (Figure 1). The most pronounced ABA accumulation occurred at 6 h in underground parts (298.97 ng/g FW, 2.33-fold increase vs. control) and at 2 h in aerial parts (294.53 ng/g FW, 1.71-fold increase). ABA predominantly accumulated in underground tissues, exhibiting an initial increase followed by decline, indicating stronger sensitivity and more sustained response to drought stress in underground organs compared to aerial tissues.

### 2.2. Differential Expression Patterns of the GuDREB Gene Family in Response to Drought Stress

Transcriptome analysis of drought-stressed *G. uralensis* tissues revealed differential expression patterns among 18 *GuDREB* genes (Figure 2, Table 1). In aerial parts, seven genes (*GuDREB1*, *GuDREB2*, *GuDREB4*, *GuDREB9*, *GuDREB14*, *GuDREB15* and *GuDREB16*) were up-regulated with distinct response kinetics: *GuDREB15* showed the earliest and strongest induction (3.07-fold at 2 h), *GuDREB4*, *GuDREB14* and *GuDREB16* peaked at 6 h, and *GuDREB1*, *GuDREB2*, *GuDREB9*, *GuDREB11* and *GuDREB12* increased at 12 h, while *GuDREB6*, *GuDREB11* and *GuDREB18* were down-regulated. In underground parts, five genes (*GuDREB3*, *GuDREB5*, *GuDREB7*, *GuDREB10* and *GuDREB13*) exhibited up-regulation, with *GuDREB5* responding most rapidly (2 h) but without significant expression change, and *GuDREB10* showing the strongest induction (1.70-fold at 6 h). Notably, *GuDREB* genes displayed tissue-specific drought responses, with no simultaneous activation in both tissues, suggesting organ-functional specialization. *GuDREB17* remained unexpressed in both tissues. These results demonstrate that *GuDREB* genes mobilize distinct regulatory networks to coordinate drought responses.

### 2.3. Genomic Scaffold Localization and Multiple Sequence Alignment of GuDREB Proteins

The 18 *GuDREB* genes were mapped to distinct genomic scaffolds (Figure 3A), indicative of uniform chromosomal distribution; multiple sequence alignment revealed two conserved domains: the YRG element (a 20-residue N-terminal basic hydrophilic region facilitating DNA binding) and the RAYD element (42–43 residues at the AP2 domain’s C-terminus regulating DRE cis-element binding affinity) (Figure 3B).

### 2.4. Physicochemical Characterization and Secondary Structure Prediction of GuDREB Proteins

Screening of the *G. uralensis* transcriptome database identified 25 putative DREB transcription factors; after removing redundant sequences, those with incomplete ORFs, and proteins containing fragmented conserved domains, 18 non-redundant *GuDREB* genes (*GuDREB1*–*GuDREB18*) were selected for further characterization (Table 2). NCBI BLAST was used to align the DREB protein sequences of *Arabidopsis thaliana* and *G. uralensis*. This alignment was validated by CDD and SMART domain analysis and confirmed that all 18 GuDREB proteins contain a single highly conserved AP2 domain. Physicochemical characterization revealed genomic lengths of 534–2864 bp, CDS lengths of 525–1509 bp, and protein lengths of 174–502 aa, with predicted molecular weights ranging from 19,521.95 to 55,106.82 Da and theoretical pI values spanning 4.97–9.32 (GuDREB2, GuDREB5, GuDREB7, GuDREB10 and GuDREB15 classified as alkaline proteins); hydrophobicity analysis showed GRAVY indices (−0.923 to −0.443, confirming hydrophilic properties with GuDREB15 as most hydrophilic), instability indices (49.74–70.74, all indicating stable proteins), and aliphatic indices (35.04–70.13). Subcellular localization predictions (WoLF PSORT) indicated nuclear localization for 14 proteins and dual nuclear/cytoplasmic localization for 4 proteins, demonstrating nuclear-centric functionality for these transcription factors and providing a foundation for functional studies of the GuDREB gene family.

To elucidate structural composition, secondary structures of GuDREB proteins were predicted (Table 3), revealing four components: random coils, α-helices, β-sheets, and extended strands. Each protein exhibited distinct secondary structure proportions: random coils predominated (highest in GuDREB13 at 57.63%, lowest in GuDREB4 at 36.25%); α-helix and extended strand contents were consistently lower than random coils, while extended strands were generally less abundant than α-helices—with the exception of GuDREB5, GuDREB17, and GuDREB18 which showed inverted ratios.

### 2.5. Phylogenetic Analysis of Plant DREB Family Members

Phylogenetic analysis of the *G. uralensis* DREB gene family with *Arabidopsis thaliana* and *Glycine max* orthologs (Figure 4) classified the 18 *GuDREB* genes into five subgroups (DREB-A1, A2, A3, A4, A6; excluding A5) according to *Arabidopsis thaliana* nomenclature: DREB-A1 comprised GuDREB5, GuDREB6, GuDREB11, GuDREB12, GuDREB14, GuDREB17 and GuDREB18; DREB-A2 contained GuDREB1, GuDREB2, GuDREB4, GuDREB7, GuDREB9 and GuDREB16; DREB-A3 solely included GuDREB15, which clustered with *Glycine max* Gm17; DREB-A4 grouped GuDREB3, GuDREB8 and GuDREB13; and DREB-A6 consisted exclusively of GuDREB10. The evolutionary tree revealed high sequence homology (>80% average identity in AP2 domains) among *G. uralensis*, *Arabidopsis thaliana*, and *Glycine max*, suggesting conserved functional roles in stress-responsive pathways.

### 2.6. Integrated Analysis of Gene Structures, Conserved Domains, and Motifs in the G. uralensis DREB Gene Family

Analysis of exon-intron structures, conserved domains, and motifs across 18 *GuDREB* genes—integrated with phylogenetic reconstruction (Figure 5)—revealed conserved and divergent genomic features. Gene structure analysis (Figure 5D) showed three nucleotide sequence components (UTR, introns, exons), where all members exclusively contained exons except *GuDREB12* (possessing UTR) and *GuDREB5*, *GuDREB7* and *GuDREB12* (containing introns). Domain analysis (Figure 5C) confirmed all 18 proteins harbor only the AP2 domain, consistent with DREB subfamily characteristics. Further motif scanning identified 12 conserved motifs (Figure 5B); Motif 1, Motif 2, Motif 3, and partial Motif 7 constitute the AP2 domain, explaining their universal presence. Motif distribution aligned with phylogenetic relationships, for instance, *GuDREB6*, *GuDREB11*, *GuDREB12*, *GuDREB14* and *GuDREB18* shared an identical motif profile comprising Motif 1, Motif 2, Motif 3, Motif 5, Motif 7, and Motif 9.

### 2.7. Analysis of Cis-Regulatory Elements in G. uralensis DREB Subfamily Members

Cis-regulatory element analysis of 2000 bp promoter regions upstream of *GuDREB* genes (Figure 6 and Figure 7) identified 17 stress-responsive elements, including 251 light-responsive elements, hormone-responsive elements (methyl jasmonate [MeJA], abscisic acid [ABA], salicylic acid [SA], gibberellin [GA], and auxin), and abiotic stress-responsive elements (anaerobic induction, defense/stress, low-temperature, and hypoxia-specific). Additional elements comprised 37 MYB-binding sites, circadian control motifs, zein metabolism regulation, meristem expression, endosperm expression, and cell cycle regulation sites. These results indicate that *GuDREB* genes mediate *G. uralensis* development and stress responses through diverse regulatory pathways.

## 3. Discussion

The synthesis, accumulation, and regulatory mechanisms of abscisic acid (ABA) in *G. uralensis* remain poorly characterized. Previous studies indicate moderate drought stress enhances accumulation of bioactive compounds (e.g., liquiritin, glycyrrhizic acid) [21]. In this study, 18 GuDREB transcription factors were identified from the *G. uralensis* transcriptome database. Research confirms that certain transcription factors function as structural components of transcriptional complexes within the nucleus [22]. Subcellular localization predictions demonstrated nuclear targeting for all 18 *GuDREB* genes, aligning with the established literature and confirming their role as nuclear proteins capable of transcriptional regulation; DREB subfamily proteins exhibit high sequence similarity and functional redundancy.

DREB proteins typically function as key regulators in abiotic stress responses such as low temperature, high temperature, drought, and salt, as well as in plant developmental regulation [23,24,25]. Within the six groups (A1–A6) of the *Arabidopsis thaliana* DREB subfamily, each exhibits distinct structural and functional characteristics; members of the A2 group, DREB2s (DREB2A/B/C), are primarily associated with the regulation of abiotic stresses including low temperature, high temperature, drought, and salt [26,27,28,29,30]. Similarly, *Zea mays* ZmDREB2A, *Glycine max* GmDREB2A, and *Broussonetia papyrifera* BpDREB2 are induced by temperature, drought, and salt stress [31,32,33]. Bioinformatics analysis revealed that all 18 *G. uralensis DREB* genes contain only a single conserved AP2 domain (Figure 5C), classifying them within the DREB subgroup of the AP2/ERF transcription factor family. Upon exposure to stresses like drought, plants utilize signal transduction to activate these transcription factors, enabling their binding to corresponding cis-acting elements. This subsequently stimulates the RNA polymerase II catalytic complex to initiate transcription of target genes. The resulting gene products then act within the organism to regulate and elicit responses to internal and external signals [34]. Phylogenetic tree analysis indicated a closer clustering relationship between *G. uralensis*, *Arabidopsis thaliana* and *soybean*. Based on the classification of the *Arabidopsis thaliana DREB* gene family into six subgroups (A1–A6), the 18 *G. uralensis DREB* genes were distributed among the A1, A2, A3, A4, and A6 subgroups (Figure 4). The DREB-A6 subgroup in *G. uralensis* contains only GuDREB10, which clustered with *Arabidopsis thaliana* RAP2.4 (AT1G22190.1, AT1G78080.1) (Figure 4). Studies demonstrate that DREB-A6 subgroup members AP2.4 and RAP2.4 respond to high salt, drought, and heat [35,36]. Transcriptome analysis in *Arabidopsis thaliana* showed that drought and salt stress treatments induced the expression of related AP2/DREB transcription factors (*RAP2.4*, *AT1G78080.1*) [37]. Lin et al. found that *Arabidopsis thaliana* overexpressing RAP2.4 exhibited enhanced drought tolerance, although water loss showed no significant difference compared to the wild type [35]. The expression level of GuDREB10 in the underground parts of *G. uralensis* was 1.70-fold higher at 6 h of drought stress compared to the 0 h group, suggesting that GuDREB10 responds to drought stress in *G. uralensis* roots. The *G. uralensis* gene *GuDREB15* clustered within the DREB-A3 subgroup, showing close phylogenetic relationships to *Arabidopsis thaliana AtERF019* (*AT1G22810.1*) and *Glycine max GmDREB8* (*Glyma.17G216100*) (Figure 4). The *soybean GmDREB8* (*Glyma.17G216100*) protein localizes to the nucleus. Heterologous expression of *GmDREB8* in transgenic *Arabidopsis thaliana* resulted in shorter roots under mannitol treatments of varying concentrations, indicating increased sensitivity to drought stress [38]. *GuDREB15*, the gene showing the fastest response to drought stress in the aboveground parts of *G. uralensis*, exhibited the most significant upregulation at 2 h, with expression levels 3.07-fold higher than the 0 h treatment group, indicating greater sensitivity to drought stress, which aligns with previous research. This suggests that *GuDREB15* plays a crucial role in responding to drought stress in the aerial parts of *G. uralensis*.

Promoter cis-element prediction analysis revealed that *DREB* genes possess hormone-responsive elements such as those for abscisic acid (ABA), methyl jasmonate (MeJA), salicylic acid (SA), gibberellin (GA), and auxin (Figure 6), along with elements for defense against stress and light response, suggesting that *GuDREB* genes may participate in ABA response and stress resistance pathways, as well as potentially in plant photoresponse processes. Studies indicate that under certain drought conditions, when plant roots sense soil drought, they produce substantial ABA; acting as a signaling molecule, ABA is transported to the aerial parts, regulating stomatal closure in leaves to enhance water retention and consequently improve plant drought resistance [39,40,41]. This study found that ABA content in *G. uralensis* under drought stress primarily accumulated in the underground parts (Figure 1), showing an overall trend of initially increasing followed by a decrease; simultaneously, it was observed that the underground parts of *G. uralensis* responded more sensitively to drought stress, while the aboveground parts exhibited a more sustained response. Given that drought stress rapidly induced the expression of DREB transcription factors, with expression levels significantly higher than the 0 h control group, it is hypothesized that genes regulating DREB transcription exist upstream. The *GuDREB15* gene contained the highest number of light-responsive elements (23) (Figure 6 and Figure 7), which may explain its significantly higher expression in aboveground tissues.

This study found that drought stress primarily induced the upregulation of *GuDREB* genes in the aboveground parts, with *GuDREB15* exhibiting the most significant upregulation, reaching 3.07-fold that of the 0 h treatment group; whereas in the underground parts, *GuDREB10* expression was significantly upregulated, with its level at 6 h of drought stress being 1.70-fold that of the 0 h group, indicating that this gene family can mobilize related genes for differential expression in response to stress under drought conditions (Figure 2, Table 1). Studies show that DNA binding, protein interactions, nuclear localization, and transcriptional activation activity are associated with the domains or amino acid motifs of transcription factors [42]. Conserved motif analysis of the genes (Figure 3B) revealed that *GuDREB15* contains two conserved YRG and RAYD motifs, similar to the structure of *DREB* genes in other species, which may account for the functional conservation of *GuDREB15*. The gene structure of *GuDREB* genes (Figure 5D) showed that *GuDREB15* is intronless, potentially because the absence of introns can reduce the time required from transcription to translation, enabling rapid gene expression and functional protein production to trigger corresponding mechanisms within the plant or respond to environmental changes [43]. Subcellular localization results indicated that the *GuDREB15* protein localizes to the nucleus (Table 2), suggesting it may play an important transcriptional regulatory role in responding to drought stress, which may also explain the greater sensitivity of the *GuDREB15* gene under drought stress. It has been found that the *GmDREB8* gene can negatively regulate plant drought resistance through osmotic adjustment and the antioxidant defense system [38], leading to the inference that the *GuDREB15* gene may also negatively regulate drought resistance in *G. uralensis*.

In total, this study identified and screened 18 *DREB* genes in *G. uralensis* and conducted bioinformatics analysis along with drought stress treatment. The physicochemical properties and secondary structures of the *G. uralensis DREB* gene family proteins were predicted. Analyses including the phylogenetic tree, promoter cis-acting elements, gene structure, and chromosomal localization of the *G. uralensis DREB* genes were performed, leading to the inference of potential biological functions for some genes. However, the specific mechanisms of action require functional validation by cloning the *G. uralensis DREB* genes and verifying their function through overexpression or suppression of expression in *G. uralensis.*

## 4. Materials and Methods

### 4.1. Plant Material and Data Sources

Plant material of *G. uralensis* was collected and taxonomically identified by our laboratory. The collection site was the Wetland Reserve in Hebukeser County, Tacheng Prefecture, Xinjiang Uygur Autonomous Region, China. The *G. uralensis* genome sequence data and corresponding annotation files were obtained from the Gur Genome Project database hosted by RIKEN: https://www.riken.jp/en/ (accessed on 2 May 2025). Protein sequences of DREB transcription factors from *Glycine max* were downloaded from the SoyBase database (https://www.soybase.org/ (accessed on 3 May 2025)). Protein sequences for the *Arabidopsis thaliana* DREB gene family were retrieved from The *Arabidopsis thaliana* Information Resource (TAIR) database (https://www.arabidopsis.org/ (accessed on 3 May 2025)).

### 4.2. Bioinformatics Analysis of the GuDREB Gene Family

Based on transcriptome sequencing data generated in this study, 18 *GuDREB* genes possessing complete open reading frames (ORFs) were identified. The deduced protein sequences of DREB transcription factors from *Arabidopsis thaliana* and *G. uralensis* were subjected to BLAST analysis using the National Center for Biotechnology Information (NCBI) service. The presence of the “AP2” domain within the *G. uralensis* protein sequences was confirmed using both the Conserved Domain Database (CDD) and SMART web servers. The physicochemical properties of the proteins encoded by the *G. uralensis DREB* gene family were predicted using the ExPASy ProtParam tool (https://web.expasy.org/protparam/ (accessed on 8 May 2025)). These properties included protein sequence length, molecular weight (MW), theoretical isoelectric point (pI), grand average of hydropathicity (GRAVY), instability index, and aliphatic index. Subcellular localization predictions were performed using WoLF PSORT (https://wolfpsort.hgc.jp/ (accessed on 9 May 2025)). Multiple sequence alignment of the *G. uralensis* DREB protein sequences was conducted using Jalview software (Vers. 2.11.2.4). Phylogenetic analysis was performed using MEGA11 software. A neighbor-joining (NJ) phylogenetic tree was constructed with default parameters. The resulting tree file (NWK format) was visualized and annotated using the Interactive Tree Of Life (iTOL) online tool (https://itol.embl.de/ (accessed on 9 May 2025)).

Conserved motifs within the *G. uralensis* DREB proteins were predicted using the MEME Suite (http://meme-suite.org/ (accessed on 10 May 2025)). Genomic location information for the *DREB* genes was extracted from the *G. uralensis* genome annotation files. The conserved motifs, domains, phylogenetic relationships, and exon/intron structures of the GuDREB proteins were integrated and visualized using TBtools software (Vers. 2.056). The genomic positions of the 18 identified *GuDREB* genes were mapped onto their respective scaffolds within the *G. uralensis* genome assembly using TBtools. Chromosomal localization visualization was achieved using the MG2C online tool (http://mg2c.iask.in/mg2c_v2.1/ (accessed on 11 May 2025)). To assess cross-species conservation, the amino acid sequences of the conserved AP2 domains from *G. uralensis* were aligned with their counterparts from *Arabidopsis thaliana* and *Glycine max*. Cis-regulatory elements within the 2000 bp promoter region upstream of the predicted transcription start site for each *GuDREB* gene were identified using the PlantCARE database and subsequently analyzed with TBtools software.

### 4.3. Plant Material Treatment

In mid-May, collected *G. uralensis* seeds were manually dehulled and sun-dried for 5 days. To break seed dormancy, the seeds were treated with 98% (*w*/*w*) concentrated sulfuric acid for 55 min. Subsequently, seeds were thoroughly rinsed with distilled water to remove residual acid. The treated seeds were sown in pots filled with vermiculite substrate. Five sowing points (hills) were made per pot, with three seeds sown per hill. Trays with a depth of 3 cm were placed beneath the pots. Plants were cultivated under controlled environmental conditions: photosynthetic photon flux density (PPFD) of 360 μmol·m^−2^·s^−1^ and a photoperiod of 16 h light/8 h dark. Upon seed germination, 500 mL of half-strength Murashige and Skoog (1/2 MS) nutrient solution was supplied to each pot every 5 days. Additional distilled water was added as needed based on substrate moisture levels. Seedlings were thinned to one plant per hill at 14 days after germination. When plants reached a height of approximately 20 cm, drought stress was simulated by irrigating the pots with MS nutrient solution containing 10% (*w*/*v*) PEG 6000. Treated plants were sampled at five time points (0, 2, 6, 12, and 24 h after stress initiation). Sampling was performed by separating the plant at the tiller node: the underground parts (roots and rhizomes) below the node and the aboveground parts (shoots) were collected separately. All samples were immediately flash-frozen in liquid nitrogen and stored at −80 °C until further analysis. These samples were used for both ultra-performance liquid chromatography-tandem mass spectrometry (UPLC-MS/MS) quantification of abscisic acid (ABA) content and transcriptome sequencing.

### 4.4. Sample Tissue Processing

Fresh tissue samples (aerial and underground parts) of *G. uralensis* were pulverized in liquid nitrogen to a fine powder. Approximately 1 g of powder (fresh weight, FW) was aliquoted into a centrifuge tube. Subsequently, 5 mL of methanol was added, and the mixture was vortexed vigorously (30 s) to achieve homogeneity. Ultrasoni-assisted extraction was performed under controlled conditions: 25 °C, 1000 W power, and 30 min duration. Following sonication, extracts were incubated at 4 °C for 30 min and centrifuged (10,000× *g*, 5 min, 4 °C). The supernatant was transferred to a new labeled tube. The residual pellet was re-extracted with 5 mL fresh methanol through repetition of the ultrasonic extraction, incubation, and centrifugation steps. Supernatants from both extractions were combined in the same labeled tube. The combined methanol extracts were stored at 4 °C for subsequent use.

### 4.5. Standard Curve Preparation

A stock solution of abscisic acid (ABA) standard was prepared at 1 mg/mL by dissolving accurately weighed ABA standard in methanol. Serial dilutions were then performed using methanol to prepare standard working solutions at concentrations of 1, 5, 10, 25, 50, 75, and 100 μg/mL.

### 4.6. Chromatographic Conditions

Chromatographic separation was performed on an ACQUITY UPLC BEH C18 column (1.7 μm, 2.1 mm × 50 mm; Waters, United States) maintained at 30 °C. The mobile phase consisted of: Eluent A: 0.1% (*v*/*v*) formic acid in water; Eluent B: Acetonitrile. The flow rate was 0.3 mL/min and the injection volume was 1.0 μL. The following gradient elution program was employed: 0.0–3.0 min: 20% B to 98% B; 3.0–4.5 min: 98% B (isocratic); 4.5–5.0 min: 98% B to 20% B; 5.0–6.0 min: 20% B (isocratic; column re-equilibration).

### 4.7. Mass Spectrometry Conditions

Analysis was performed using electrospray ionization (ESI) under the following optimized source parameters: Ion source temperature: 150 °C; Capillary voltage: 3.0 kV; Desolvation gas temperature: 450 °C; Desolvation gas flow: 800 L/h; Cone gas flow: 150 L/h; Nebulizer gas flow: 7.0 Pa. Data acquisition utilized multiple reaction monitoring (MRM) mode. The specific MRM transitions and associated instrument parameters for abscisic acid (ABA) are detailed in Table 4. Representative MRM chromatograms for ABA are presented in Figure 8.

### 4.8. Linearity Analysis for Abscisic Acid Quantification

The prepared ABA standard solutions were analyzed under the chromatographic and mass spectrometric conditions detailed in Section 2.6 and Section 2.7. A calibration curve was constructed by plotting the peak area (*y*-axis) against the standard concentration (*x*-axis). As summarized in Table 5, ABA exhibited excellent linearity (R^2^ > 0.99) across the tested concentration range.

### 4.9. RNA Extraction and Transcriptome Sequencing

This part of the work was performed by Shanghai Majorbio Bio-pharm Technology Co., Ltd. (Shanghai, China). The sequencing results were compared and analyzed with the *G. uralensis* genome (https://www.riken.jp/en/, accessed on 2 May 2025).

### 4.10. Validation of RNA-Seq Data Using qRT-PCR

To validate the differentially expressed genes obtained from RNA sequencing, the researchers randomly selected four *G. uralensis DREB* genes for qRT-PCR verification. The experiment was performed using gene-specific primers on a LightCycler^®^ 480 Real-Time PCR System (Roche, Basel, Switzerland), with the lectin gene found in licorice serving as the internal control (Table 6). Relative expression calculated by 2^−ΔΔCt^ method.

### 4.11. Statistical Analysis

Data analysis was performed using GraphPad Prism 8.0. Statistical significance between treatment groups was determined by one-way analysis of variance (ANOVA) followed by Tukey’s post hoc test. Significance thresholds were defined as follows: *p* < 0.05: statistically significant; *p* < 0.01: highly statistically significant. All experiments included three independent biological replicates. All data collection and calculations were conducted with 3 biological replicates.

## 5. Conclusions

Currently, cultivated *G. uralensis* serves as the primary source of commercial *G. uralensis*, and the quality of cultivated *G. uralensis* has consistently been one of the main factors restricting its development; consequently, enhancing the quality of cultivated *G. uralensis* remains a current research hotspot. Moderate abiotic stress can facilitate the synthesis and accumulation of active ingredients in *G. uralensis*. This study determined the accumulation characteristics of abscisic acid (ABA) in the aboveground and underground parts of *G. uralensis* under drought stress, as well as the expression characteristics of *G. uralensis DREB* genes under such stress. It investigated the relationship between the expression of the GuDREB gene family and ABA biosynthesis in *G. uralensis* and their potential roles, laying a theoretical foundation for further exploration of *GuDREB* gene functions and potentially providing new perspectives for *G. uralensis* resistance breeding. This research contributes to revealing the mechanism by which *G. uralensis* regulates the accumulation of bioactive components through ABA under drought conditions and provides a scientific basis for improving *G. uralensis* quality.

## Figures and Tables

**Figure 1 ijms-26-09235-f001:**
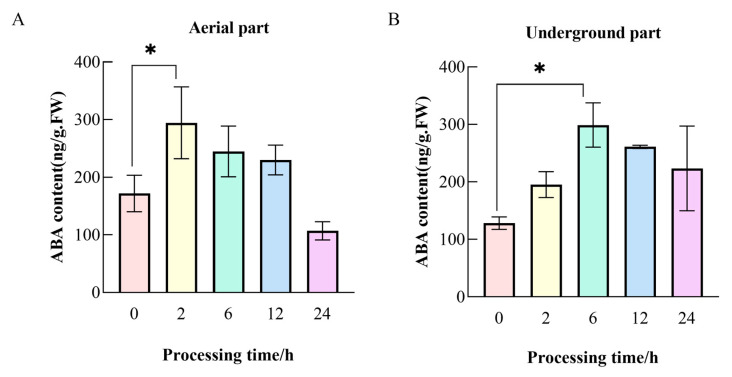
ABA content in aerial and underground tissues of *G. uralensis* under drought stress at different time points. Note: (**A**) Aerial part of *G. uralensis*; (**B**) Underground part of *G. uralensis*; The figure presents the results of multiple comparisons between all time points; only statistically significant or highly significant differences are indicated; * indicates that the difference between the treatment group and the control group (0 h group) is significant (*p* < 0.05).

**Figure 2 ijms-26-09235-f002:**
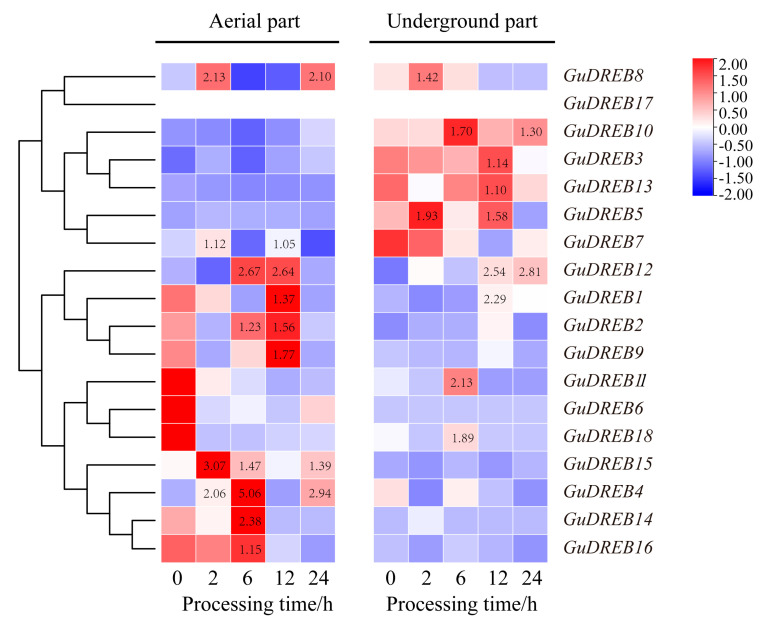
Heatmap of 18 *DREB* gene expression patterns in aerial and underground tissues of *G. uralensis* under abiotic stress at different time points. Note: The value in the heat map represents the up−regulation multiple of the expression level of the gene relative to 0 h.

**Figure 3 ijms-26-09235-f003:**
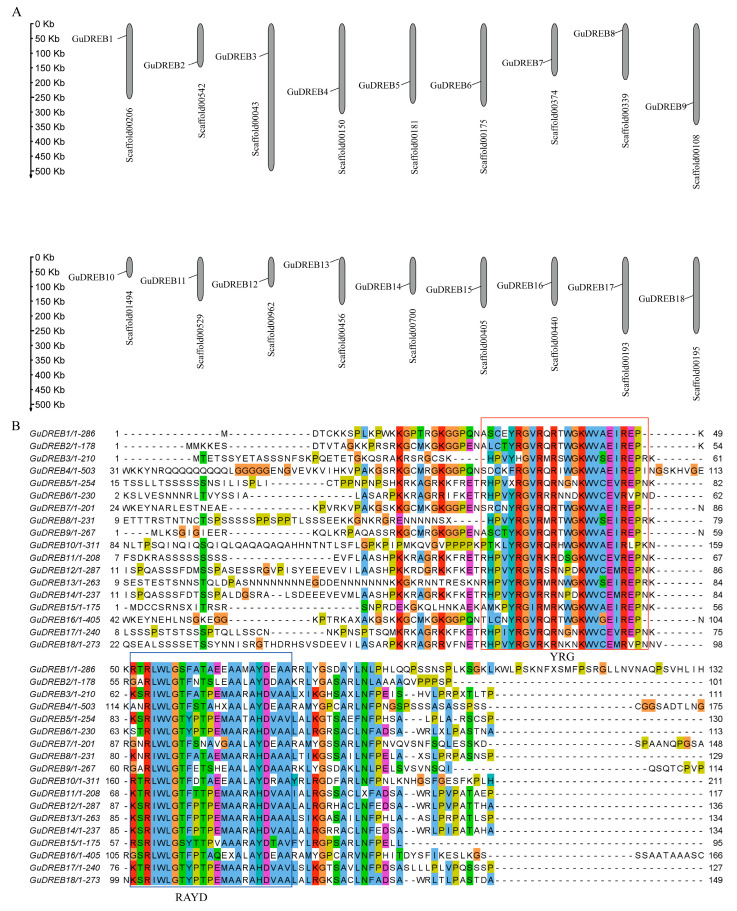
Genomic localization and protein sequence alignment of 18 GuDREB genes in *G. uralensis*. Note: (**A**) Scaffold mapping of GuDREB genes; (**B**) Multiple protein sequence alignment with the YRG domain highlighted in red and the RAYD domain in blue.

**Figure 4 ijms-26-09235-f004:**
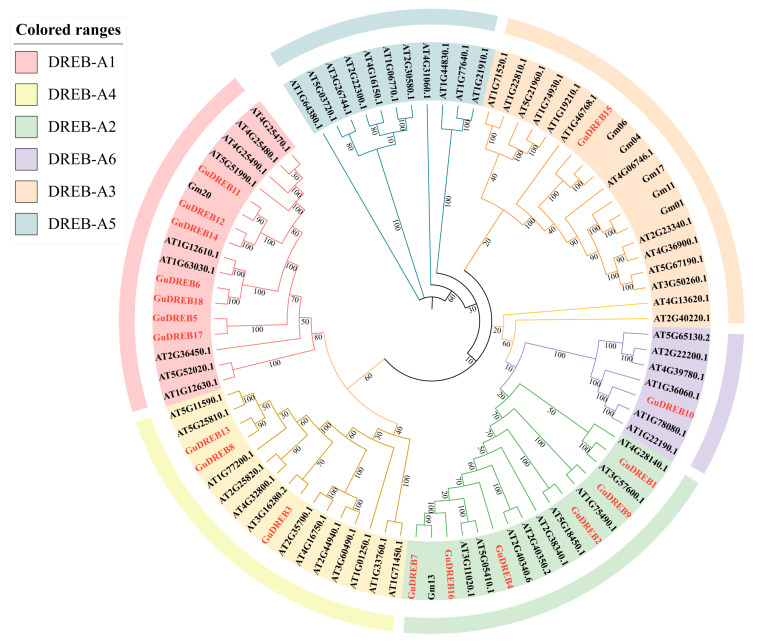
Phylogenetic tree analysis of the DREB gene family in *G. uralensis*, *Arabidopsis thaliana*, and *Glycine max*.

**Figure 5 ijms-26-09235-f005:**
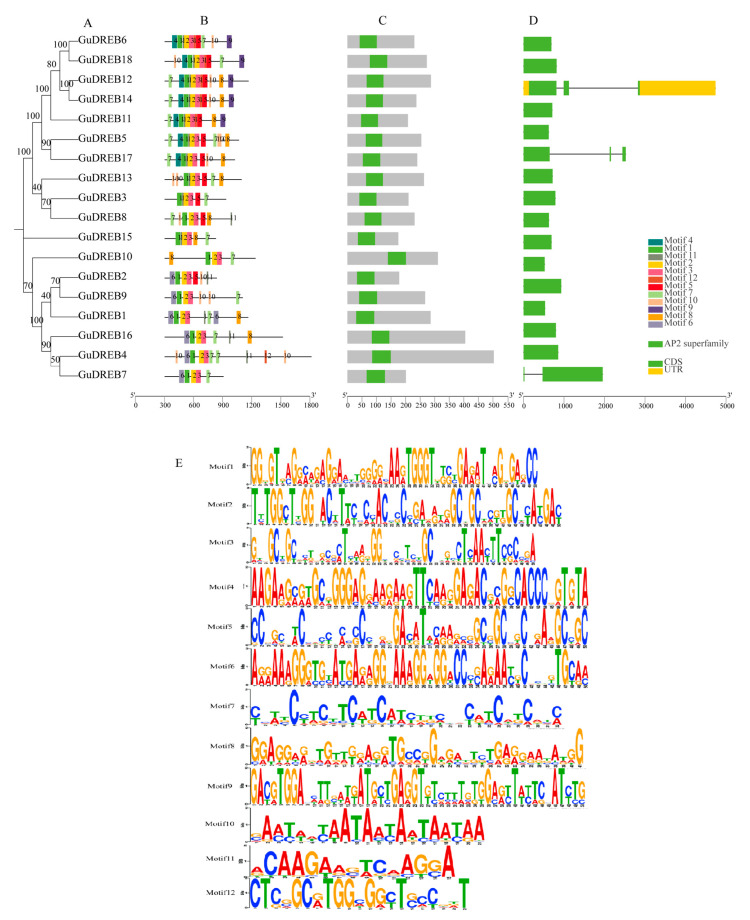
Comprehensive Analysis of the *G. uralensis* DREB Gene Family. Note: (**A**) Phylogenetic tree of protein sequences; (**B**) Conserved motif distribution; (**C**) Domain architecture; (**D**) Exon-intron structure; (**E**) Motif sequence logos.

**Figure 6 ijms-26-09235-f006:**
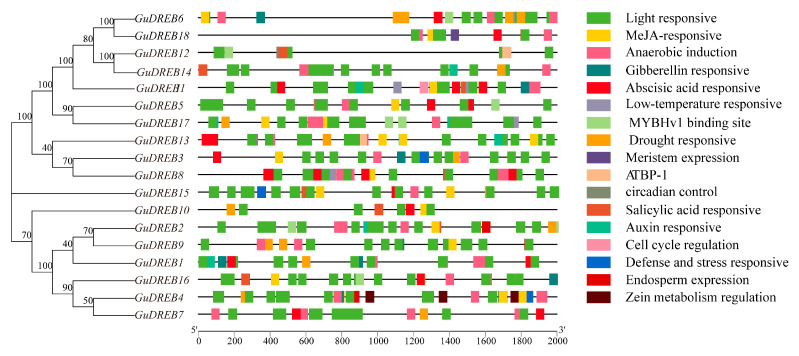
Cis-Regulatory element prediction in promoter regions of 18 *DREB* genes from *G. uralensis*.

**Figure 7 ijms-26-09235-f007:**
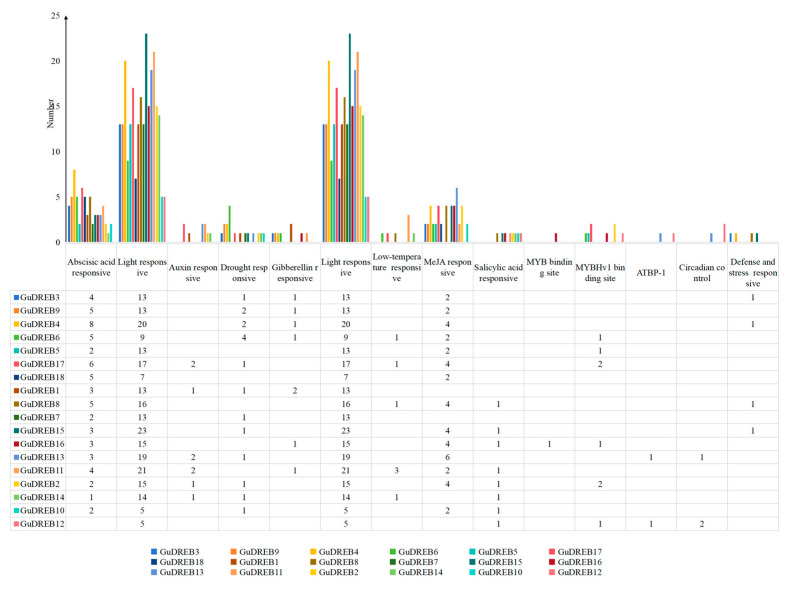
Statistics on thenumber of keycis-acting elementsin thepromoter sequences of 18 *G. uralensis* DREB genes.

**Figure 8 ijms-26-09235-f008:**
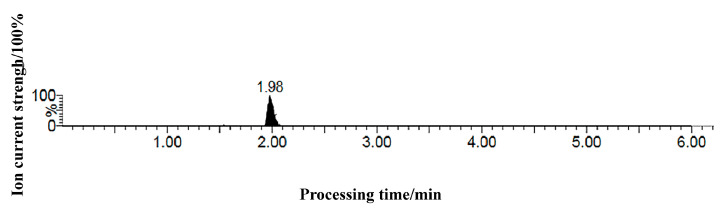
MRM chromatogram of Abscisic acid standard.

**Table 1 ijms-26-09235-t001:** Expression levels (TPM) of 18 *GuDREB* genes in aerial and underground tissues of *G. uralensis* under abiotic stress at different time points.

Gene Name	Aerial Part					Underground Part				
	0 h	2 h	6 h	12 h	24 h	0 h	2 h	6 h	12 h	24 h
* GuDREB1 *	0.89 ± 0.12	0.56 ± 0.08	0.15 ± 0.03	1.22 ± 0.15	0.16 ± 0.02	0.21 ± 0.04	0.09 ± 0.01	0.14 ± 0.02	0.48 ± 0.06	0.43 ± 0.05
*GuDREB2*	0.48 ± 0.07	0.08 ± 0.01	0.59 ± 0.08	0.75 ± 0.09	0.13 ± 0.02	0.00 ± 0.00	0.07 ± 0.01	0.07 ± 0.01	0.27 ± 0.03	0.00 ± 0.00
*GuDREB3*	4.70 ± 0.62	8.78 ± 1.12	4.01 ± 0.52	8.02 ± 1.03	10.49 ± 1.35	23.53 ± 3.06	21.89 ± 2.84	19.96 ± 2.59	26.92 ± 3.50	13.89 ± 1.81
*GuDREB4*	0.16 ± 0.02	0.33 ± 0.04	0.81 ± 0.10	0.13 ± 0.02	0.47 ± 0.06	0.36 ± 0.05	0.09 ± 0.01	0.33 ± 0.04	0.19 ± 0.02	0.11 ± 0.01
*GuDREB5*	0.00 ± 0.00	0.07 ± 0.01	0.04 ± 0.01	0.04 ± 0.01	0.00 ± 0.00	0.57 ± 0.07	1.10 ± 0.14	0.39 ± 0.05	0.90 ± 0.12	0.00 ± 0.00
*GuDREB6*	2.46 ± 0.32	0.10 ± 0.01	0.26 ± 0.03	0.00 ± 0.00	0.66 ± 0.08	0.00 ± 0.00	0.00 ± 0.00	0.00 ± 0.00	0.00 ± 0.00	0.00 ± 0.00
*GuDREB7*	20.31 ± 2.64	22.80 ± 2.96	16.98 ± 2.21	21.39 ± 2.78	16.22 ± 2.11	28.85 ± 3.75	27.27 ± 3.55	22.67 ± 2.95	18.76 ± 2.44	22.44 ± 2.92
*GuDREB8*	4.46 ± 0.58	9.50 ± 1.24	1.44 ± 0.19	2.01 ± 0.26	9.38 ± 1.22	6.48 ± 0.84	9.23 ± 1.20	6.64 ± 0.87	4.23 ± 0.55	4.26 ± 0.56
*GuDREB9*	0.77 ± 0.10	0.00 ± 0.00	0.47 ± 0.06	1.36 ± 0.18	0.00 ± 0.00	0.10 ± 0.01	0.06 ± 0.01	0.04 ± 0.01	0.27 ± 0.03	0.00 ± 0.00
* GuDREB10 *	28.90 ± 3.76	26.76 ± 3.48	17.16 ± 2.23	27.77 ± 3.61	44.55 ± 5.79	65.70 ± 8.54	64.83 ± 8.43	111.64 ± 14.51	76.35 ± 9.93	85.10 ± 11.06
*GuDREB11*	8.21 ± 1.07	2.75 ± 0.36	1.61 ± 0.21	0.66 ± 0.09	0.97 ± 0.13	1.86 ± 0.24	1.17 ± 0.15	5.04 ± 0.65	0.37 ± 0.05	0.39 ± 0.05
*GuDREB12*	0.39 ± 0.05	0.22 ± 0.03	1.04 ± 0.13	1.03 ± 0.13	0.37 ± 0.05	0.26 ± 0.03	0.58 ± 0.07	0.43 ± 0.05	0.66 ± 0.08	0.73 ± 0.09
*GuDREB13*	1.41 ± 0.18	0.78 ± 0.10	0.00 ± 0.00	0.27 ± 0.03	0.48 ± 0.06	13.97 ± 1.82	5.70 ± 0.74	12.54 ± 1.63	15.38 ± 2.00	8.08 ± 1.05
*GuDREB14*	0.26 ± 0.03	0.13 ± 0.02	0.62 ± 0.08	0.00 ± 0.00	0.00 ± 0.00	0.00 ± 0.00	0.08 ± 0.01	0.00 ± 0.00	0.00 ± 0.00	0.00 ± 0.00
* GuDREB15 *	76.38 ± 9.93	234.19 ± 30.44	112.25 ± 14.60	65.57 ± 8.52	106.33 ± 13.82	26.10 ± 3.40	16.09 ± 2.10	33.96 ± 4.41	17.35 ± 2.26	32.92 ± 4.28
* GuDREB16 *	19.04 ± 2.48	16.86 ± 2.20	21.96 ± 2.85	5.36 ± 0.70	1.78 ± 0.23	4.12 ± 0.54	1.85 ± 0.24	4.81 ± 0.62	3.39 ± 0.44	1.38 ± 0.18
*GuDREB17*	0.00 ± 0.00	0.00 ± 0.00	0.00 ± 0.00	0.00 ± 0.00	0.00 ± 0.00	0.00 ± 0.00	0.00 ± 0.00	0.00 ± 0.00	0.00 ± 0.00	0.00 ± 0.00
* GuDREB18 *	3.33 ± 0.43	0.00 ± 0.00	0.00 ± 0.00	0.13 ± 0.02	0.16 ± 0.02	0.45 ± 0.06	0.03 ± 0.01	0.85 ± 0.11	0.06 ± 0.01	0.03 ± 0.01

Note: Data are presented as mean ± standard deviation (SD).

**Table 2 ijms-26-09235-t002:** Physicochemical properties of DREB transcription factor family proteins in *G. uralensis*.

Gene Name	Genomic (bp)	CDS (bp)	Aliphatic Index	Average Hydrophilicity	Length (aa)	pI	Mw (Da)	Sublocation	Instability Index (II)
* GuDREB1 *	858	858	65.75	−0.860	285	6.47	32,149.59	nucl	53.48
*GuDREB2*	534	534	60.79	−0.672	177	8.44	19,521.95	cyto, nucl	35.04
*GuDREB3*	630	630	68.18	−0.445	209	5.38	23,142.96	nucl	63.95
*GuDREB4*	1509	1509	58.86	−0.897	502	6.64	55,106.82	nucl	53.82
*GuDREB5*	762	762	59.41	−0.841	253	8.61	28,264.57	nucl	47.20
*GuDREB6*	690	690	70.74	−0.507	229	5.32	25,572.11	cyto, nucl	56.38
* GuDREB7 *	1158	603	55.20	−0.816	200	9.30	21,799.72	cyto, nucl	43.40
* GuDREB8 *	693	693	49.74	−0.715	230	5.16	24,845.95	nucl	65.91
* GuDREB9 *	801	801	69.70	−0.443	266	5.13	29,171.77	cyto, nucl	52.94
* GuDREB10 *	933	933	63.94	−0.666	310	7.03	34,350.45	nucl	52.27
*GuDREB11*	624	624	55.85	−0.543	207	5.68	22,747.49	nucl	60.17
*GuDREB12*	2864	861	58.04	−0.698	286	6.12	32,225.61	nucl	64.33
*GuDREB13*	789	789	56.68	−0.774	262	5.59	28,224.83	nucl	53.50
*GuDREB14*	711	711	58.39	−0.766	236	5.37	26,557.51	nucl	65.84
* GuDREB15 *	525	525	60.11	−0.923	174	9.32	19,567.12	nucl	51.94
*GuDREB16*	1925	1215	55.57	−0.713	404	4.97	43,894.36	nucl	38.60
*GuDREB17*	720	720	64.06	−0.628	239	5.12	26,568.61	nucl	52.37
* GuDREB18 *	819	819	64.96	−0.626	272	5.21	30,668.00	nucl	70.13

**Table 3 ijms-26-09235-t003:** Predicted secondary structures of GuDREB proteins.

Gene Name	α-Helix	Extended Strand	β-Turn	Random Coil
* GuDREB1 *	29.47%	16.84%	9.47%	44.21%
*GuDREB2*	35.03%	13.56%	9.04%	42.37%
*GuDREB3*	29.67%	15.31%	5.74%	49.28%
*GuDREB4*	44.02%	12.55%	7.17%	36.25%
*GuDREB5*	15.02%	22.92%	7.11%	54.94%
*GuDREB6*	29.26%	25.33%	4.80%	40.61%
* GuDREB7 *	37.00%	13.50%	6.00%	43.50%
* GuDREB8 *	22.17%	21.30%	4.35%	52.17%
* GuDREB9 *	26.69%	18.05%	6.77%	48.50%
* GuDREB10 *	23.55%	15.48%	6.45%	54.52%
*GuDREB11*	31.88%	17.87%	7.73%	42.51%
* GuDREB12 *	28.32%	19.58%	5.94%	46.15%
* GuDREB13 *	24.81%	12.60%	4.96%	57.63%
*GuDREB14*	33.05%	15.25%	5.93%	45.76%
* GuDREB15 *	33.91%	14.94%	4.02%	47.13%
*GuDREB16*	33.17%	12.87%	10.15%	43.81%
*GuDREB17*	23.85%	25.52%	7.11%	43.51%
* GuDREB18 *	24.63%	25.00%	8.46%	41.91%

**Table 4 ijms-26-09235-t004:** MRM parameters for quantification of abscisic acid (ABA) in *G. uralensis*.

Compounds	Retention Time (min)	Relative Molecular Weight/Da	Ionization Mode	MS (*m*/*z*)	MS2 (*m*/*z*)	Cone- Voltage/V	Collision Energy/eV
Abscisic acid	2.00	264	ESI-	263.2	153.0 *	22	12

Note: There are two daughter ions with high response efficiency in glycyrrhiza abscisic acid determined by the liquid mass spectrometry system, where * represents the highest response efficiency and is used as the quantitative ion.

**Table 5 ijms-26-09235-t005:** Linear parameters for the determination of Abscisic acid contents.

Component	Linear Equation	R^2^	Linear Range (ng·mL^−1^)
Abscisic acid	Y = 103.72X − 99.05	0.9956	1.2~100

**Table 6 ijms-26-09235-t006:** Primers used for *GuDREB* gene expression analysis.

Primer Name	Primer Sequences F (5′-3′)	Primer Sequences R (5′-3′)
* GuDREB3 *	GCAGCAAGCACCCAGTTTAC	CTGCCATTTCAGGGGTAGCA
* GuDREB * *7*	GAAAGGGAAAGGAGGACCCG	TGTTCGGCTCCCGAATTTCA
* GuDREB * *8*	AACCATCAAAGGCTCCTCCG	ACGAACTCGCTATTCGGGTC
* GuDREB * *15*	AGTTGGTGCCAGAGTCGATG	AGGTCAAGGCAATCTTCGGG
*Gu* *Lectin*	CTGATGCAGAGCTTCAAATCGAG	TTCGGAAGGAAGGTTGAGGTAAG

## Data Availability

All data are presented in the article and the Appendix A.

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
