# Peer review of "Bioinformatics Analysis and Expression Profiling Under Abiotic Stress of the DREB Gene Family in *Glycyrrhiza uralensis"

_ijms, 2025, doi:10.3390/ijms26189235_

Round 1
Reviewer 1 Report
Comments and Suggestions for Authors
This manuscript focuses on the identification and functional analysis of 18 DREB transcription factors in Glycyrrhiza uralensis. Based on transcriptomic data, the authors characterized these genes through sequence analysis, phylogenetic comparison, promoter element prediction, and expression profiling under drought stress. The study aims to elucidate potential roles of GuDREB genes in abiotic stress responses, especially drought, and to contribute to stress-resistance gene selection in this medicinally important species.
While the study addresses a relevant topic and includes comprehensive analyses, several issues concerning structure, data presentation, and scientific rigor must be addressed before the manuscript can be considered for publication.
Major and Minor Issues
- Abstract Format and Scientific Naming:
The first sentence of the abstract uses only the species epithet “uralensis”, which is a serious formatting issue. The full binomial name Glycyrrhiza uralensis must be written in its first appearance in the abstract to avoid ambiguity. This is a basic scientific writing requirement and should be carefully checked throughout the manuscript. - Structural Logic of the Manuscript:
The current organization of the Results section is confusing. Sections 2.1 and 2.2 (expression and hormone response analysis) appear too early. A more logical structure would be:
- Gene identification and classification
- Phylogenetic and promoter analysis
- Chromosomal location
- Followed by expression profiles and physiological responses.
I recommend reordering these sections to reflect the typical progression from gene identification to functional inference.
- Statistical Analysis in Figure 1 Unclear:
It is not clear how the significance was calculated in Figure 1. Was the statistical test done only between 0 h and 2 h? Or was 0 h used as a control for multiple comparisons across all time points (6 h, 12 h, 24 h)? The figure legend and Results text do not provide this information. Moreover, statistical comparisons for 6 h/12 h/24 h are missing, which limits interpretability. This must be clarified, and appropriate statistical results should be shown or described. - Expression Pattern Interpretation Lacks Statistical Support:
The authors claim that several genes (GuDREB1, GuDREB2, GuDREB4, GuDREB9, GuDREB14, GuDREB15, and GuDREB16) are upregulated in aerial parts. However, these conclusions are not convincingly supported by statistical data. There is no table or supplementary material showing fold-change values or significance levels. For example, for GuDREB14, expression increases from 0.26 to 0.62 (at 6 h), and for GuDREB16, from 19.04 to 21.96. These are modest changes and likely not significant without statistical analysis. Please provide original expression values with p-values or adjusted p-values to support any upregulation claims. Table 1 alone is insufficient. - Chromosomal Mapping Reliability (Figure 3A):
All 18 GuDREB genes are mapped to scaffolds rather than chromosomes. This raises concern about the genome assembly quality used. Glycyrrhiza uralensis reportedly has 8 chromosomes; are none of the DREB genes anchored to them? The manuscript should explain whether a high-quality genome is available and, if not, discuss the limitations this poses. Please confirm the reliability of chromosomal mapping results. - Phylogenetic Tree (Figure 4) Visualization and Methodology:
The bootstrap support values are indicated by circles of varying size, which are difficult to interpret. It is strongly recommended to use numerical bootstrap values instead of visual symbols. Furthermore, some branches appear to have a bootstrap of 0, which questions the reliability of the tree.
In addition, the use of MEGA11 software for phylogenetic tree construction is not cited. Please include the proper reference for MEGA11, as software tools must be cited like all other resources. - RNA-seq Validation via qRT-PCR (Figure S1) Lacks Significance Testing:
Figure S1, which presumably validates RNA-seq data through qRT-PCR, does not show any statistical testing (e.g., error bars, significance levels). Without this, the consistency between RNA-seq and qPCR data is not convincing. Significance tests must be performed and clearly marked in the figure. - Formatting – Species Name Usage Throughout the Manuscript:
It is a basic formatting requirement that the species name be written in full (Glycyrrhiza uralensis) at first mention, and abbreviated thereafter (e.g., G. uralensis). Please revise the entire manuscript accordingly.
Recommendation:
Major revision. The manuscript presents potentially valuable findings, but numerous issues in structure, interpretation, data validation, and formatting must be addressed before further consideration.
Reviewer 2 Report
Comments and Suggestions for Authors
Glycyrrhiza is a medicinal plant with a wide spectrum of action. Therefore, the problem posed in this manuscript is relevant. To solve the problem, the authors used a variety of methods, with the help of which the problem was successfully solved.
The authors of the submitted manuscript used a standard set of methods to solve the problem. Using a combination of molecular and bioinformatic methods allows us to obtain reliable results and draw appropriate conclusions. I am a supporter of the idea that in order to draw a reliable conclusion, several methods should be used. When only bioinformatic methods are used, the conclusions depend on the database, parameters, and assumptions used. But when these conclusions are confirmed by molecular and biochemical experiments, they become more reliable. Therefore, I believe that the submitted manuscript meets the requirements of the journal.
There are minor comments
15 - the name Glycyrrhiza is missing
Table 1 - provide data with statistical deviation
Table 4 and Figure 8 should be moved either to the Results section or to the Supplementary
Round 2
Reviewer 1 Report
Comments and Suggestions for Authors
I have carefully reviewed the revised manuscript [Manuscript ID: ijms-3812617] and examined the authors’ point-by-point responses to my previous comments. The authors have addressed all major concerns raised in the first review, including corrections to the abstract and scientific naming, clarification of statistical analyses, reorganization and logical flow of the Results section, improved figure clarity (including numerical bootstrap values for the phylogenetic tree), presentation of qRT-PCR validation with significance testing, and consistent formatting throughout the manuscript.
The authors have also provided clear explanations and supporting revisions for methodological and data presentation issues, which significantly improve the clarity, rigor, and interpretability of the study. Overall, the manuscript now meets the scientific standards for publication, and all critical issues from the initial review have been satisfactorily resolved.